# Safety of COVID-19 Vaccines: Spotlight on Neurological Complications

**DOI:** 10.3390/life12091338

**Published:** 2022-08-29

**Authors:** Giacomo Tondo, Eleonora Virgilio, Andrea Naldi, Angelo Bianchi, Cristoforo Comi

**Affiliations:** 1Neurology Unit, Department of Translational Medicine, S. Andrea Hospital, University of Piemonte Orientale, 13100 Vercelli, Italy; 2Neurology Unit, Department of Translational Medicine, Maggiore della Carità Hospital, University of Piemonte Orientale, 28100 Novara, Italy; 3Neurology Unit, San Giovanni Bosco Hospital, 10154 Turin, Italy; 4Department of Neuroscience “Rita Levi Montalcini”, University of Turin, 10126 Turin, Italy

**Keywords:** neurological adverse effects, SARS-CoV-2, vaccination, Guillain-Barré syndrome, Bell’s palsy, Vaccine-Induced Thrombotic Thrombocytopenia, myelitis, multiple sclerosis

## Abstract

The COVID-19 pandemic has led to unprecedented demand on the global healthcare system. Remarkably, at the end of 2021, COVID-19 vaccines received approvals for human use in several countries worldwide. Since then, a solid base for response in the fight against the virus has been placed. COVID-19 vaccines have been shown to be safe and effective drugs. Nevertheless, all kinds of vaccines may be associated with the possible appearance of neurological complications, and COVID-19 vaccines are not free from neurological side effects. Neurological complications of COVID-19 vaccination are usually mild, short-duration, and self-limiting. However, severe and unexpected post-vaccination complications are rare but possible events. They include the Guillain-Barré syndrome, facial palsy, other neuropathies, encephalitis, meningitis, myelitis, autoimmune disorders, and cerebrovascular events. The fear of severe or fatal neurological complications fed the “vaccine hesitancy” phenomenon, posing a vital communication challenge between the scientific community and public opinion. This review aims to collect and discuss the frequency, management, and outcome of reported neurological complications of COVID-19 vaccines after eighteen months of the World Health Organization’s approval of COVID-19 vaccination, providing an overview of safety and concerns related to the most potent weapon against the SARS-CoV-2.

## 1. Introduction

The novel coronavirus SARS-CoV-2 causes a pulmonary-systemic coronavirus disease (COVID-19) which was firstly described in Wuhan, China, in December 2019. Fastly, COVID-19 became a worldwide health emergency, and, in March 2020, COVID-19 was declared a pandemic [1]. At the end of 2021, COVID-19 vaccines have been released for widespread vaccination, receiving approvals in several countries for human use and providing a solid hope in the battle against the virus [2]. Starting from the 31st December 2020, four major vaccine types have been used with an extensive distribution worldwide: the BNT162b2 (Pfizer–BioNTech), the mRNA1273 (Moderna), the ChAdOx1 nCov-19 (Oxford–AstraZeneca), and the Ad26.COV2.S (Janssen) vaccine [3]. The first two represent COVID-19 mRNA-based vaccines encoding the spike protein antigen of SARS-CoV-2; the other two vaccines are recombinant adenoviral vectors. The 2021 trials, including these four vaccines, reported high effectiveness ranging from 66.6 to 95% at preventing COVID-19 infections, while injection-site injury, pain and reactions, axillary lymphadenopathy, fatigue, myalgia, arthralgia, and headache were reported as the most frequent adverse effects [3]. Even if in the clinical trials the safety records were satisfying, since the beginning of the global vaccination program, vaccine hesitancy, mainly due to safety concerns, represented a challenge. Worries raised due to reported serious or even fatal events, such as anaphylaxis, myocarditis, and thrombotic events, primarily associated with the adenovirus-vectored vaccines [4]. However, the subsequent revisions of the UK Medicines and Healthcare Products Regulatory Agency and the European Medicines Agency concluded that the rate of fatal thrombotic events in the vaccinated population was non-significant compared with the general population, confirming a favorable risk/benefit for the vaccination [5]. Despite the reported COVID-19 vaccine acceptance rate of over 60% of the population in most countries, skepticism and concerns remain, and neurological adverse effects represent a particularly feared outcome [6]. Neurological complications of COVID-19 vaccines include central and peripheral nervous system manifestations, ranging from minimal and tolerable adverse effects to life-threatening toxicity. 

Neurological side effects of SARS-CoV-2 vaccines are frequent but primarily non-serious [7]. They can be classified as mild or severe neurological events [8]. Mild or non-serious neurological events include weakness, muscle and joint pain, transient sensory symptoms, and headache [7]. These adverse reactions are the most frequently described adverse effects in the majority of reports. They usually occur acutely and are expected to be transient [7,9,10]. Minor adverse effects are generally more severe after the second dose than the first dose and seem to hit women more than men [11,12]. In the nationwide descriptive study of García-Grimshaw and colleagues, the overall incidence of non-serious neurological effects of COVID-19 vaccination with the Pfizer–BioNTech vaccine was 600.7 cases per 100,000 administered doses, and a headache was the most frequent disturbance [7]. 

Neurological severe adverse effects after COVID-19 vaccination are infrequent events [7]. Still, they are the main reason to create vaccination hesitancy. They include the Guillain-Barré syndrome (GBS), seizures and syncope, encephalitis, meningitis, myelitis, demyelinating disorders, myasthenic disorders, thrombocytopenia, cerebrovascular events, facial nerve palsy and other cranial nerve neuropathies [8,13]. The increased risk of cerebral venous sinus thrombosis (CVST), which initially was associated with the Oxford–AstraZeneca vaccine, is an example of a rare and severe adverse neurological effect. Another fearful neurological complication already reported in the Oxford–AstraZeneca clinical trial is transverse myelitis, diagnosed in two cases in the treatment arm [13]. GBS, which can result in life-threatening sequelae, has received particular attention, arousing concerns by the US Food and Drug Administration (FDA) due to the suspected link with the adenoviral vector vaccines [8]. Overall, vaccinations with vector-based vaccines seem burdened by more frequent or severe side effects than the mRNA vaccines [14,15]. While viral vector vaccine usually induces a strong immune response, thus potentially associated with a higher rate of adverse effects, and especially systemic reactions, mRNA vaccine is a novel vaccine platform with good immunogenicity and safety, showing a lower rate of complications, usually local side effects [16,17]. Nevertheless, the COVID-19 vaccine is generally well-tolerated, adverse effects are usually self-limiting, and there is no clear evidence of a higher rate of neurological disorders associated with the COVID-19 vaccination [9]. Conversely, it is not the first time in human history that vaccinations have been associated with developing neurological disorders. In people vaccinated against the pandemic influenza A (H1N1), a relatively increased risk for Bell’s palsy and paresthesia was reported; however, the risk for other neurological complications such as GBS was found to be similar to that of unvaccinated people [18]. In the recent past, severe neurologic events have been speculated with the measles-mumps-rubella vaccinations. However, the association with several postulated complications, including encephalitis, aseptic meningitis, and autism, has been rebutted [19]. This narrative review aimed to collect and discuss literature data regarding neurological complications of SARS-CoV-2 vaccines after eighteen months of the World Health Organization’s approval of COVID-19 vaccination. Here we provide an overview of the frequency, distribution, management, and outcome of neurological complications of COVID-19 vaccines. The last literature search was done on 30 June 2022.

## 2. Headache

Headache is one of the most frequent adverse effects of the COVID-19 vaccine, reported in approximately half of the vaccine recipients [20,21]. It usually occurs within 72 h after the vaccination. It resolves within hours or a few days later, manifesting as a single episode of moderate-to-severe intensity in most cases [22]. Pain characteristics are variable, but the headache is often bilateral, involving frontal and temporal regions, and is associated with fatigue and myalgia. The cumulative prevalence of all headache episodes, from mild to severe, is slightly higher after the second dose than in the first [22,23].

Headache occurs with any of the FDA and European Medicines Agency-approved vaccines; an Italian report describes the higher risk of developing a headache after the Oxford–AstraZeneca vaccine, followed by the Pfizer–BioNTech [24]. A history of headache, both migraine, and non-migraine types, is associated with a higher risk of suffering from headache after COVID-19 vaccination than controls without pre-existing headache [23]. A delayed presentation of the headache, occurring around one week after immunization, should be carefully evaluated. It has been associated with CVST in patients administered an adenovirus vector-based COVID-19 vaccine [25].

## 3. Cerebrovascular Events

It is unclear whether several vaccines may trigger cerebrovascular events [26]. As a fact, stroke occurrence has also been reported after the COVID-19 vaccination. Neurovascular thrombosis may involve both the arterial and venous systems, and different pathogenic mechanisms have been proposed. Ischemic and hemorrhagic strokes are reported, but—although rare—the most described event is CVST, which may occur as isolated or in association with the so call Vaccine-Induced Thrombotic Thrombocytopenia (VITT).

### 3.1. Vaccine-Induced Thrombotic Thrombocytopenia and Cerebral Venous Sinus Thrombosis

VITT is probably the most severe neurological complication of COVID-19 vaccines and is mostly—but not solely—related to adenoviral ones, particularly Oxford–AstraZeneca [27,28]. Cases associated with the Moderna vaccine are anecdotal, and no definite conclusions can be drawn [29,30,31]. VITT consists of immune-mediated thrombocytopenia with thrombosis with onset usually within 2–3 weeks after the first dose of vaccination [32]. The term VITT was coined because of its similarity with heparin-induced thrombocytopenia (HIT), which is caused by the emergence of antibodies activating platelets after heparin administration [33,34]. In this syndrome, heparin becomes immunogenic by binding the platelet factor 4 (PF4), thus inducing antibody formation [35]. As a result, platelets are activated, and thrombocytopenia develops, leading to potential thrombosis, predominantly in the deep veins system or in pulmonary embolism [36]. Unlike HIT, patients with VITT do not have previous exposure to heparin. However, similarly to HIT, high levels of antibodies directed to PF4-polyanion complexes are detectable. The mechanisms by which vaccines promote the development of antibodies are still unclear. Several pathogenic hypotheses have been proposed, and both vaccines and host factors seem to be involved. The PF4 binding to adenovirus vectors and pro-inflammatory and immunogenic signals are the most reported [37,38,39].

These IgG autoantibodies cause platelet activation via the FcyIIa receptor, stimulating the immune response (activation of monocytes, neutrophils, and endothelial cells) and platelet consumptions, leading to increased risk of thrombosis [40].

The effects of these interactions may be detected by laboratory tests, and high D-dimer levels, positive antibodies against PF4 (detected by enzyme-linked immunosorbent assay, ELISA), and low platelet count are considered the hallmark for serological diagnosis [4,41]. Compared with HIT, thrombosis in patients with VITT may occur at unusual sites, including CVST, splanchnic, portal, or hepatic veins [42,43]. Reasons for this specificity are still unknown, and research is ongoing to clarify the question [44]. Thrombosis in cerebral veins may be massive, thus explaining the potential catastrophic evolution of VITT [43,45]. Brain ischemia, intracerebral hemorrhage (ICH), and subarachnoid hemorrhage may complicate CVST, requiring a prompt intervention [46,47]. Neurological symptoms recall those of CVST and depend on the territories involved in the thrombosis. The clinical onset is often insidious with malaise and progressive worsening headache, not responding to analgesic treatments. Focal neurological symptoms, seizures, vomiting, blurred vision, and consciousness impairment may suddenly develop. Jointly with other systemic symptoms, abdominal, back, and chest pain may coexist, reflecting thrombosis in other sites [48]. 

Due to the severity of the condition, prompt recognition of the syndrome is crucial to timely and correctly contrast thrombocytopenia and thrombosis. Whether the fatality rate was initially very high, mortality has significantly decreased over time, explaining the improvement in the early identification and intervention [49]. Several—constantly evolving—recommendations released from expert consensus and international societies are available for VITT management [50,51,52]. The hallmark of therapies includes the administration of high-dose intravenous immunoglobulin and non-heparin anticoagulation, depending on platelet count, clinical status, and residual organ function of the patient [41,52,53,54]. Some authors reported successful mechanical thrombectomy for massive CVST [45,55]. Even if knowledge is limited regarding the indications and timing of endovascular procedure in CVST, it may be considered an option in this life-threatening condition. Overall, management of VITT requires a multidisciplinary approach at centers with neuro-interventional and surgical experience [56]. VITT is a rare adverse event after COVID-19 adenoviral vaccines. The incidence seems to range from 1/125,000 to 1 in 1 million vaccinated cases [30,41]. A recent metanalysis estimated a higher incidence of VITT, reaching 28/100,000 incidence, and most frequently presented with CVT following deep vein thrombosis/pulmonary thromboembolism and splanchnic vein thrombosis, and about one-third of patients had a fatal outcome [43]. Risk factors are still unknown, although female and younger age have been potentially identified in the first descriptions [57].

CVTS may also occur outside the context of VITT [28,58,59,60,61]. Non-VITT CVST patients seem to be older than VITT-associated, with a lower number of veins thrombosed at first diagnosis and a lower rate of superimposed ICH. As a consequence, outcomes seemed to be more favorable [62]. Without any hematologic disorder associated with CVST vaccine-related, management does not differ from standard cases.

Several studies tried to identify a distinctive pattern of CVST-related VITT compared to non-VITT CVST. They found higher mortality or dependency after discharge and a shorter time interval between vaccination and clinical onset in the CVST-VITT group. In addition, CVST-VITT was more often complicated by ICH, explaining the worst outcome for these patients [63]. Interestingly, CVST with thrombocytopenia was almost exclusively described after vector-based vaccination, reported in 57% of cases of CVST following the Oxford-AstraZeneca vaccine [64].

### 3.2. Ischemic and Hemorrhagic Strokes

Ischemic strokes have been reported after the COVID-19 vaccination, and several descriptions are available [5,65,66]. The association with adenoviral vaccines seems more frequent, with a clinical onset within three weeks since the inoculation [67]. Both lacunar infarction and large vessel occlusion are reported, mainly involving the middle cerebral artery and its vascular territories [5,67]. Women below 60 years old seem more affected [67]. Management of acute ischemic stroke does not differ from routine clinical practice, and reperfusion therapies are indicated within international guidelines and recommendations. Both intravenous thrombolysis (IVT) and mechanical thrombectomy (MT) have been performed in ischemic stroke patients after Oxford–AstraZeneca and Pfizer–BioNTech vaccinations, with variable outcomes [5,65]. Cerebral arterial thrombosis has also been reported jointly with CVST or in the context of VITT [68,69,70,71,72]. In these cases, treatments for acute ischemic strokes have to be weighted considering the neuro-imaging findings and laboratory test results, particularly for low platelet counts that may contraindicate IVT [73,74]. Overall, arterial thrombosis causing stroke seems much less frequent than venous events [75].

Hemorrhagic events include ICH and SAH. Isolated ICH outside any known coagulation disorder is rarely reported [76,77]. In these cases, uncertainties regarding the causal relationship between vaccination and cerebral bleeding remain. Instead, to date, there is no description of isolated SAH after COVID-19 vaccination. Conversely, ICH and SAH are frequently described as a complication of CVST, as previously mentioned. Therefore, they are mostly reported after adenoviral vaccines, particularly Oxford–AstraZeneca, in people below 60 years old and within two weeks after the vaccination [67]. Management of ICH and SAH follows the general standard of care indications. The use of tranexamic acid is reported, as well as neurosurgical procedures in case of severe ICH, including hematoma drainage, external ventricular drain, and decompressive craniectomy [67,76]. However, aggregate data indicate that cerebrovascular events are not increased after COVID-19 mRNA vaccines, as reported in several series [78,79,80]. The mechanisms of ischemic and hemorrhagic stroke related to COVID-19 vaccines are still unclear, and, at least in part, they may overlap with those of VITT. Hypercoagulable states related to the vaccine’s inflammatory process may promote clots formation and cause strokes [81]. Beyond the factors previously described for VITT, alterations in protein S levels, thrombocytopenia, hypofibrinogenemia, folate deficiency with elevated homocysteine, factor XIII deficiency, and antiphospholipid antibodies have been detected in the context of strokes after COVID-19 vaccination [39]. In addition, both vaccine components and host factors may produce an autoimmune response, leading to thrombosis. 

## 4. Multiple Sclerosis

Multiple sclerosis is a chronic autoimmune disease of CNS, recognizing a genetic background on which several environmental factors, mainly infections, can act as a disease trigger [82,83]. In the past, vaccines (such as hepatitis B, yellow fever vaccination, and human papillomavirus) were awarded as a possible determinant for MS onset. However, several studies have failed to show an association [84]. Currently, only live attenuated vaccines are generally contraindicated in MS patients, particularly those receiving immunosuppressants. Live attenuated vaccines have the potential to cause an infection if the vaccine is administered during treatment with an immunosuppressant, the ability of the immune response of the MS patient to clear the infection could be impaired, possibly resulting in a worsening of MS symptoms. Nowadays, most international and national consensus from MS experts recommend COVID-19 vaccination, and mRNA vaccines are considered safe [84,85].

We found several studies reporting the new onset of MS after SARS-Cov2 immunization [86,87,88,89,90,91,92,93]. All cases were relapsing-remitting MS. Demographic and clinical characteristics are comparable to the general MS population ranging from optic neuritis (ON) to transverse myelitis (TM) or brainstem syndrome. The neurological manifestation presented from one day after the first vaccine dose to several weeks after the second immunization showing a temporal variability [89,91]. The case described by Halva et al. displayed a familiarity with MS [87]. The majority showed intrathecal immunoglobulin synthesis. The systematic review by Ismail et al. considered twelve MS-like presentations (both new-onset and relapses) correlated to the COVID19 vaccine reported up to September 2021. Only one patient among the new-onset was not vaccinated with mRNA [88]. The majority of the described cases of MS after the COVID-19 vaccine already presented high dissemination in space (DIS) and time (DIT) at diagnosis (i.e., previous history of neurological symptoms, concomitant enhancing and non-enhancing lesions, black holes lesions in T1 MRI); therefore, we could assume that most probably they already had the autoimmune disease long before the vaccine administration. Indeed, MS can sometimes be diagnosed in preclinical phases (the so-called radiologically isolated syndrome—RIS), demonstrating that the disease is pathologically and radiologically present long before the clinical manifestations. Even though only one case of the presented had a previous history of the clinically isolated syndrome (ON with unremarkable previous brain MRI) [91], several of the reported patients already showed a high DIS at diagnosis [86,87,91]. Furthermore, real-world data do not corroborate the hypothesis of a higher risk of new MS onset or MS relapse associated with the COVID-19 vaccination [91,94,95,96].

## 5. Neuromyelitis Optica Spectrum Disorder and MOG Antibody Disease

Neuromyelitis Optica Spectrum disorder (NMOSD) is a demyelinating disease that frequently presents with extensive longitudinal TM (LETM) and monolater or bilateral ON. The illness pathogenesis recognizes the presence of antibodies against aquaporin-4 (Abs-AQP4) in serum and CSF, and the disease is primarily idiopathic [97]. NMOSD is often in differential diagnosis with MOG antibody disease (MOGAD), displaying very different histopathology and prognosis [98]. NMOSD and MOGAD are, therefore, antibodies-mediated autoimmune syndromes. Seven cases of AQP4-NMOSD were reported, one with inactivated vaccine, one with a viral vector, and four with mRNA vaccines [99,100,101,102]. Latency ranged from one day after the first dose to eighteen days after. One patient presented with area postrema and hypothalamic syndrome. Five patients showed signs of TM (one with short TM, one with short TM and brainstem syndrome, and three with LETM). Two Abs-AQP4 monolateral ON were also reported after mRNA vaccine [103,104]. One case of Abs-AQP4 LETM NMOSD associated with Sjogren’s disease after 18 days from Pfizer–BioNTech vaccination was described in a 64-year-old male with no medical history [88,89]. 

Less commonly, MOG antibodies positive syndromes were described [105,106]. Two LETM after Oxford–AstraZeneca vaccination [105,106], one with incomplete recovery after steroid and plasma exchange were reported [106]. 

For NMOSD and MOGAD post-vaccination, the outcome was favorable for the majority of the patients. Little information about long-term prognosis or treatment is at the moment available.

## 6. Myelitis 

Post-vaccination acute TM cases have been reported following vaccination against hepatitis B, diphtheria, tetanus, and influenza. However, the pathogenesis remains unclear. It is assumed that recombinant or live-attenuated viruses can induce autoimmunity through molecular mimicry or epitopes spreading mechanisms. Several cases have been described of TM after COVID-19 immunization with different types of vaccine. A review by Garg et al. reports that adenoviral vector-based COVID-19 vaccines are more frequently associated with the spreading of TM [8].

Three TM cases were reported in the four RCT Oxford–AstraZeneca. They occurred on days 10, 14, and 64 after vaccination, respectively [107]. However, only one patient was considered possibly related to the vaccine, with an independent neurological committee considering the most likely diagnosis to be an idiopathic, short segment TM. Conversely, two additional cases were later determined unrelated to immunization based on the previous history of undiagnosed MS or extreme latency of the event [107,108]. In addition, national vigilance boards registered 45 cases of myelitis in the United Kingdom and nine patients in Germany after the Oxford–AstraZeneca vaccination [106,109]. More recently, other case reports all over the world suggested cases of TM after the Oxford–AstraZeneca vaccine [109,110,111,112,113,114,115]: some authors described cervical TM, other dorsal TM, mostly short TM but also LETM ranging from day fourafter the first dose to day fourteen. Veggezzi et al. suggested a correlation, although specifying that the definition of acute vaccine-induced myelitis according to the WHO criteria remained unclear [116]. To note, the majority of patients had an unremarkable medical history, and usually, a favorable outcome occurred with treatments except for the case of Notghi et al.: a 58-year-old man with a history of pulmonary sarcoidosis developed seven days after vaccination, a T2-T10 LETM and deteriorated even after steroids showing an extension up to C1. Neurosarcoidosis was ruled out with thorax CT and CT PET. The patient finally improved with plasma exchange sessions [114]. We may speculate that the severity of the presentation might be related to an autoimmune predisposition of the subject and the extension of the neurological event (LETM rather than TM). 

On the other hand, in stage III clinical trial for Pfizer–BioNTech, 18,860 patients were vaccinated, and four (<0.01%) developed serious adverse events related to the vaccine: one patient reported transient leg paresthesia) [117]. He was not diagnosed with TM, and it is unclear if any MRI or CSF analysis was undertaken. However, real-world data on MS, MOGAD, NMOSD, and TM populations (963 patients), exposed to the Pfizer–BioNTech vaccine has been reassuring. Vaccine-associated new or worsening neurological symptoms occurred in less than ~15% of patients, commonly early in the post-vaccination period (within the first week) and mostly self-resolving within two weeks [95,96]. Other single cases of TM after Pfizer–BioNTech vaccine and after Moderna could be found in the literature. Cases are from all over the world, both occurring in females and males, mostly of young age, but TM in 69-, 75-, 81-, and 85-year old-patients were also described [118,119,120,121,122,123,124]. Symptoms onset ranged from three days from the first dose to four days after the second immunization, and even though long-term data are missing outcome was usually good except for very old patients or patients with LETM. Two cases after the Moderna vaccine were also described by Fitzsimmons et al. and Gao et al.: a 63-year-old male who developed symptoms of TM 17 h after the second dose of an mRNA vaccine (Moderna) and a 76-year-old female with a cervical gd+ LETM developed six days after Moderna vaccine [125]. However, he presented low vitamin B12, which could have influenced the presentation. A case series of CNS demyelinating disease after an mRNA vaccine was recently published. It included one case of a Caucasian man diagnosed with TM [89].

Tahir et al. reported a case of a 44-year-old woman presenting with C3-T1 LETM after receiving the Janssen immunization ten days before presentation. Interestingly while undergoing plasma exchange, she developed Bell’s palsy. A second MRI after plasma exchange showed an improvement [126].

## 7. Optic Neuritis

ON is an isolated inflammation of the optic nerve in its course (anterior, retrobulbar, chiasmatic, i.e.,). It could represent a manifestation of MS, NMOSD, or recognize metabolic or infectious etiology, but ON may be monophasic and isolated in fewer cases. We found eleven idiopathic ON after COVID-19 immunization, the majority being females, with a good outcome. 

The first case of left-isolated ON one week after a single dose of the Janssen vaccine and a prompt steroid response was reported. MOG Abs were not tested, and isoelectrofocusing showed intrathecal synthesis indicating a chronic inflammatory process. Therefore, strict follow-up of the patients is mandatory to intercept a possible evolution in clinically defined MS or MOGAD [127].

Barone et al. reported that two patients aged 48 and 31 years presented with acute optic neuritis a few days (twelve and nine) after the first dose of mRNA vaccination. The first patient had a typical presentation of unilateral ON and was treated with steroids with partial recovery. MRI was unremarkable. The second patient initially had a transient loss of vision after exposure to high temperature (defined by the Authors as Uhthoff’s phenomenon), followed by persistent monocular dyschromatopsia and central scotoma diagnosed with ON. No information regarding CSF of both patients, MRI, or the outcome of the second patient is reported [128]. Four ON in Germany following vaccination with Pfizer–BioNTech were reported in the study of Kaulen et al. [122]. In Koh hospital-based study, covering a four-month period during which 1,398,074 people received at least one dose of COVID-19 mRNA vaccine, 457 patients with a spectrum of neurological disorders were recorded. Two female patients (0.4%) developed ON (48 and 62 years). One was idiopathic, occurring thirty-three days after the second dose. However, the second one developed an ON one day after the first vaccine dose and was later diagnosed with NMOSD AQP4+ Abs [103]. The authors categorized the likelihood of vaccination-associated or a concurrent and coincidental illness using the WHO Adverse Events Following Immunization (AEFI) causality assessment, Brighton criteria framework (Certain, Probable, Possible, Unlikely, Unrelated) and concluded the two ON to be likely coincidental and classified as “unlikely” [103]. 

Arnao et al. reported the first case of a bilateral retrobulbar ON in a healthy woman after exposition to the first dose of Oxford–AstraZeneca vaccine [129]. Antibodies tested negative. After steroids, she fully recovered. 

Finally, a retrospective study by Assiri et al. described 18 patients referred to or presented to the Saudi German Hospitals in Jeddah, Saudi Arabia, with neurological complications from March to September 2021 after receiving a COVID-19 vaccine. Ten patients received the Oxford–AstraZeneca vaccine, and eight the Pfizer–BioNTech vaccine [5]. Of the 18 patients, 2 presented with ON (one bilateral and one monolater) fourteen and nineteen days after vaccination. Both recovered completely. 

## 8. Myasthenia Gravis

Few cases of new-onset myasthenia gravis (MG) post-immunization have been described since March 22. All of these reports concern patients with late-onset MG AchR positive. 

First, an 82-year-old presented with bulbar symptoms. He was admitted four weeks after receiving the first dose and two days after receiving the second dose of the Pfizer–BioNTech vaccine. His symptoms started a few weeks before, occurring in the evenings, often during dinner [130]. He was later diagnosed with late-onset MG AchR positive based on immunological evaluation and neurophysiology. Despite symptomatic treatment, two months later, the patient showed an evolution in generalized MG, requiring hospitalization complicated by pneumonia, requiring ventilator support, and PEG tube insertion. He later recovered. New-onset MGs following COVID-19 vaccination are rarely reported in the literature. A review of vaccine-related adverse events in a large population found only two cases of new-onset MG following the COVID-19 vaccine, with onset within 28 days from vaccination [93]. The two reported cases involved patients in their 70s and occurred 1 to 7 days after administering the second dose of the Pfizer–BioNTech vaccine [93]. Both occurred after the second dose of Pfizer–BioNTech vaccine, one being severe, both male from Israel. 

Galassi et al. reported the first case of ocular MG after Oxford–AstraZeneca, onset eight days after the first dose. A 73-year-old male presented with painless left-sided ptosis. He was confirmed MG based on EMG and serum titer of anti-AChR antibodies on day 20 after vaccine injection [131]. To note, he showed high rheumatoid factor titer without signs of joints or arthritic involvement and a recent history of psoriasis. 

A large population-based study of more than 32 million people investigated the neurological adverse events associated with the Oxford–AstraZeneca and Pfizer–BioNTech vaccines. They found an increased risk of hospital admission for myasthenic disorders (15–21 days) in those who received the Oxford–AstraZeneca vaccine [13].

## 9. Acute Disseminated Encephalomyelitis and Weston–Hurst Syndrome

ADEM is a rare acute monophasic demyelinating autoimmune event of CNS potentially occurring after an immunological trigger, such as infections or vaccinations, and sometimes presenting with hyperacute hemorrhagic components (the so-called Weston–Hurst syndrome) [132]. Post-vaccination ADEM seems to occur at any age, with males more affected than females [133]. Several case reports and case series of post-COVID-19 infection and COVID-19 vaccination were described [134]. In contrast with ADEM, Weston–Hurst normally exhibits a very poor prognosis. 

In particular, ADEM following Oxford–AstraZeneca all occurred in middle-aged patients, with symptom onset ranging from 10 days after vaccination to a few weeks with progressive MRI and clinical improvement at follow-up [133,134,135,136]. Recovery was observed in all patients except for a case occurring in Australia [135] in a 63-year-old man with a medical history of type II diabetes mellitus, ischemic heart disease, and atrial fibrillation, who presented with progressively declining cognition, disorientation, and impaired attention. On day four after admission, he suddenly became poorly responsive. MRI showed numerous bilateral (>20) cerebral white matter lesions with both periventricular and juxtacortical involvement. Stereotactic right frontal craniotomy and open biopsy of a right frontal lesion were performed on day nine, confirming the ADEM diagnosis. He later died on day twenty of admission. Further post-mortem investigations were made, and the cause of death was registered as “ADEM in the setting of recent AstraZeneca COVID-19 vaccination” [135]. 

Regarding ADEM after mRNA vaccines, except for the case of Kania et al. (describing a 19-year-old female), all other cases occurred in patients in their forties or fifties or even in elderly patients [8,33,137,138,139,140]. In all cases, MRI and CSF findings were highly variables. The 19-year-old was injected two weeks prior with Moderna vaccine [141]. 

Reports of potential vaccine-related adverse effects from the European Medicines Agency’s (EMA) EudraVigilance database by November 2021 revealed slightly higher but altogether comparable incidences of 0.07 (Oxford–AstraZeneca) and 0.05 (Janssen) per 100,000 people per year for potential vector-based vaccine-associated ADEM, compared to 0.02 (Pfizer–BioNTech) and 0.04 (Moderna) for mRNA-vaccines [142]. 

As for Weston–Hurst, Ancu et al. [142] reported three cases with onset within nine days after the first shot of the Oxford–AstraZeneca vaccine. All the patients were treated with steroids, and two with plasma exchange with severe sequelae. One patient, despite treatments, still presented with a vegetative status a few weeks later the onset; a 25-year-old woman developed a medullary thoracic syndrome nine days after the vaccine. Spinal MRI confirmed an autoimmune acute LETM gd+ with a hemorrhagic component (a spinal variant of Weston–Hurst). Brain MRI showed bi-hemispheric white matter lesions with focal contrast enhancement. After treatment, a clinical improvement of the sensory symptoms was observed but persistent paraplegia on a six-week follow-up. Finally, a 55-year-old woman developed progressive nausea, dizziness, and meningism nine days after the first shot. She neurologically deteriorated to a severe spastic tetraparesis and coma. Brain MRI revealed multiple diffuse FLAIR hyperintense and hemorrhagic lesions. She developed hydrocephalus needing an emergency right-sided decompressive hemicraniectomy. The brain cortex biopsy from the affected right temporal lobe revealed perivascular predominantly granulocytic infiltrate and hemorrhages, confirming the diagnosis. The patient died after a few weeks after. This cases series highlighted the heterogeneity in clinical findings and the verity of outcomes in Weston–Hurst syndrome.

## 10. Autoimmune Encephalitis

Autoimmune encephalitis (AIE) is a rare, recently described group of neurological diseases associated with specific autoantibodies or classified in specific syndromes (e.g., limbic encephalitis). Various subgroups of AIE are distinguished based on autoantibodies, which may lead to clear clinical presentations and different prognoses, particularly concerning paraneoplastic and non-paraneoplastic AIE. Among them, anti-leucine-rich glioma inactivated 1 (LGI1) encephalitis is characterized by cognitive impairment, rapid progressive dementia, psychiatric disorders, faciobrachial dystonic seizures (FBDS), and refractory hyponatremia. Rarely, cerebellitis has been described after vaccination [143,144]. Zlotnik et al. reported a 48-year-old man presenting with severe fatigue developed a few days following his second dose of Pfizer–BioNTech mRNA vaccination and rapidly evolving in progressive cognitive decline and hyponatremia 2.5 weeks after. He was later diagnosed with anti-LGI1 AIE. Other authors reported series of possible AIE according to Grauss Criteria related to prior Oxford–AstraZeneca or Pfizer–BioNTech vaccination. However, none showed specific CSF or serum autoantibodies or hidden neoplastic conditions despite extensive workup [145,146]. A case of acute encephalopathy occurring in a 32-year-old Asian male within a day of receiving the first dose of Moderna was also reported [147]. The second case of acute encephalitis, myoclonus, and Sweet syndrome following the Moderna vaccine occurred in the USA [148]. Kaulen et al. also reported a case of limbic encephalitis where MRI demonstrated bilateral hippocampal hyperintensities, and CSF revealed mild lymphocytic pleocytosis (13/μL), oligoclonal bands type 2, and mild disruption of the blood-brain barrier [122]. In addition, unexplained acute encephalopathy cases have also been described after receiving the COVID-19 vaccine [149,150] and hyperacute reversible encephalopathy related to cytokine storm [151].

Up to June 2021, a total of 79 encephalitis cases were reported with the Oxford–AstraZeneca vaccine (in 99.3 million doses with a resulting incidence of almost 0.08 per 100,000). Only 20 cases in 110.6 million doses with a consequent incidence of nearly 0.02 per 100,000 were reported for Pfizer–BioNTech [145]. However, those reported data included AIE, limbic encephalitis, viral encephalitis, ADEM, Bickerstaff, and other non-infective encephalitides.

## 11. Aseptic Meningitis

Aseptic meningitis (AM) is an inflammatory disorder of the meninges that can be of iatrogenic origins (e.g., immunoglobulins or NSAID) as a complication of vaccination (against mumps, measles, rubella, and influenza). It has been previously described [152]. However, the etiology remains unclear, although a possible explanation is a reaction to vaccine adjuvants recently implied in an autoimmune/inflammatory syndrome (ASIA) [92]. All reported cases so far occurred after the RNAm vaccine, and the Pfizer–BioNTech vaccine’s adjuvant is a nanoparticle-based polyethylene glycol (PEG) stabilizer that has been implied as a trigger of ASIA syndrome in other organs [153]. AM is well recognized as a reaction to certain drugs and is frequently associated with systemic autoimmune disorders. However, Satai’s AM was negative for serum anti-PEG antibodies [154]. All cases showed unremarkable brain MRI and good response to steroids. Some also presented with fever and the majority displayed CSF pleocytosis. The first published case was in a middle-aged Japanese nurse who developed a refractory headache, fever, and signs of meningeal involvement one week after vaccination. Although the patient was assuming NSAIDs for a sporadic migraine at home, she was diagnosed with aseptic meningitis based on CSF and peripheral workup. She was initially treated with acyclovir (until viral antibody titers were negative) and later with intravenous methylprednisolone [154]. Subsequently, three other cases were reported in France [155] and Singapore after [156], all after Pfizer–BioNTech.

## 12. Guillain-Barré Syndrome

GBS is an acquired, inflammatory, acute polyradiculoneuropathy generally clinically characterized by a rapidly progressive ascending flaccid paralysis. It is a rare neurological disorder, with a rate of 1 to 2 per 100,000 person-years [157]. The term GBS includes several variants: the acute inflammatory demyelinating polyradiculoneuropathy (AIDP), which is the most common and primarily expresses demyelinating features; the acute axonal motor neuropathy, less common and characterized by a worse prognosis; the acute motor-sensory axonal polyneuropathy, with both motor and sensory involvement; the Miller Fisher syndrome (MFS), consisting of ophthalmoplegia, ataxia, and areflexia; the Bickerstaff’s brain stem encephalitis, which is considered a variant of the MFS and characterized by altered consciousness, paradoxical hyperreflexia, ataxia, and ophthalmoparesis; other less common presentations, include the pharyngeal-cervical-brachial motor variant, the pure facial diplegia and the pandysautonomic variant [158,159]. The pathogenesis of GBS is primarily immuno-mediated by an immune response triggered by preceding infections or vaccinations throughout a cross-reactivity involving the axonal or myelin constituents of the peripheral nerves [160].

The relationship between COVID-19 and GBS is highly complex since COVID-19 infection can trigger GBS [161], and COVID-19 vaccination has been associated with the development of GBS, irrespective of the vaccine used [162]. 

Most cases of GBS are described following the first dose of any vaccination [163,164,165]. The interval between vaccination and GBS onset should be within six weeks [164]. GBS after COVID-19 vaccination may occur in an extensive range of ages between 20 and 90 years [166], and the onset is usually within the second week after vaccination [166], ranging from three to 30 days [122,164]. Neurophysiological findings are often typical, with the demyelinating forms of AIDP as the most described [164,166,167]. Cranial involvement is not uncommon and facial diplegia is frequent [164,168,169,170,171]. Specifically, facial palsy as a feature of GBS at a higher rate than expected seems to be more frequent after adenovirus-vectored vaccines [167,172]. Allen and colleagues presented four cases of bifacial weakness without any other typical sign, thus not showing areflexia, objective sensorimotor signals in the limbs, or dysautonomia, all showing albumin-cytological dissociation [169]. In other case series, albumin-cytologic dissociation has been detected between 75 and 90% [164,166]. In the work of Kim and colleagues, a relatively low rate of Anti-ganglioside antibodies has been reported [164].

Prognosis may be variable. Respiratory failure was described in 30% of patients in the Kim and colleagues report and 85% of patients described by Maramattom and colleagues [168]. Treatment generally includes intravenous immunoglobulin and, in fewer patients, plasmapheresis [162,166,167]. A prompt treatment with intravenous immunoglobulin followed by early physical therapy usually induces improvement of the motor and sensory deficits [122,173], but recovery may be partial [162]. Even if recurrency of GBS after COVID-19 vaccines in patients with a history of a previous GBS has also been reported [163], a history of GBS does not increase the risk of a relapse. The Israelian study of Shapiro and colleagues on a cohort of about seven hundred patients with a previous diagnosis of GBS showed the recurrence of GBS only in one patient that was successfully treated with plasmapheresis with residual minor proximal weakness [174].

MFS is also associated with COVID-19 infection and after COVID-19 vaccination [175,176]. Ophthalmoplegia and diplopia represent the most common features in the described cases, followed by ataxia [176]. Most patients were treated with intravenous immunoglobulins and showed a favorable prognosis, with recovery within four-to-six weeks [177,178,179,180]. 

In conclusion, GBS may follow COVID-19 vaccination with any vaccine. However, the reporting rate of GBS after the Janssen vaccine analyzed from the United States Vaccine Adverse Event Reporting System (VAERS) was higher than after the Pfizer–BioNTech and the Moderna vaccines [181]. In the VAERS, a national passive surveillance system for monitoring vaccine safety, 130 cases of GBS were reported from February 2021 to July 2021, with an absolute rate increase of 6.36 per 100,000 person-years [182]. For this reason, in December 2021, the Advisory Committee on Immunization Practices recommended mRNA vaccines over the Janssen vaccine due to the latter choice’s lower benefit/risk balance [183]. Similarly, the analysis of the English National Immunisation Database of COVID-19 vaccination between 1 December 2020 and 31 May 2021 showed a rate of GBS following the Oxford–AstraZeneca significantly higher than the background rates within 28 days of the first dose [13].

A recent cohort study analyzing data from the United States Vaccine Safety Datalink assessed the incidence of GBS following COVID-19 vaccination with the Pfizer–BioNTech, the Moderna, and the Janssen vaccine. The analysis involved more than ten million participants. It confirmed a higher rate of GBS after the Janssen vaccine than the mRNA vaccines, reaching an unadjusted incidence rate of 32.4 GBS per 100,000 person-year [184]. Post-vaccination surveillance is still ongoing, and these results impose caution but need further confirmation.

## 13. Bell’s Palsy and Cranial Neuropathies

Peripheral facial nerve (Bell) palsy occurrence has been associated with physical stress, pregnancy, cancer, infections, and vaccination [185]. Since the COVID-19 vaccination trials, this neurological condition has been reported as a possible adverse effect. In the Pfizer–BioNTech clinical trial, four cases of Bell’s palsy were described in the vaccine group and no cases in the placebo arm. Similarly, in the Moderna trial, three cases of Bell’s palsy were observed in the vaccine group and one in the placebo arm [186]. Since then, great attention has been paid to this aspect. In several studies, Bell’s palsy has been observed after COVID-19 vaccination at a frequency higher than expected [186,187]. In addition, compared with other viral vaccination, facial paralysis was reported significantly more frequently after COVID-19 vaccination than other vaccines, with a higher risk for males and older individuals, specifically over 65-year-old [188]. However, other reports observed no association between facial nerve palsy and the vaccination status [189,190], calling for further surveillance in larger cohorts after vaccination. 

On the other hand, despite confirming a significant association between the administration of mRNA COVID-19 vaccines and the reporting of Bell’s palsy, the analysis of Sato and colleagues on VAERS data from the US FDA suggested that the incidence of facial palsy after COVID-19 vaccination is comparable to that for influenza vaccines [191]. 

No differences between classical Bell’s palsy and the COVID-19-associated vaccine palsy seem to be present, with an excellent response to early corticosteroid therapy [187,189]. Thus, the facial palsy outcome is generally favorable when isolated and not associated with other conditions, which should be considered in the diagnostic workup; in Patone and colleagues’ work, 6% of the cohort suffering from Bell’s palsy had a concurrent suspect diagnosis of cerebral infarction [13].

Since Bell’s palsy is considered a rare complication with a high recovery rate, COVID-19 vaccination benefits outweigh the possible link with its occurrence. In addition, a study reported a higher-than-expected rate of Bell’s palsy after SARS-CoV-2 infection, which the vaccine should prevent [192]. 

Olfactory dysfunction, the most frequent neurological complication after COVID-19, can also be a rare adverse effect of COVID-19 vaccine administration [193]. Lechien and colleagues reported six cases of post-COVID-19 vaccination olfactory and gustatory disturbances, which recovered within seven weeks, and none of the patients reported long-term disorders [194]. A rare case of phantosmia after the COVID-19 vaccine showing MRI evidence of enhancement of the olfactory bulbs and tracts has been described [195].

In isolated cases, mRNA vaccination has been described to be associated with sixth and fourth cranial nerve palsy [196]. Cases of trigeminal neuralgia have also been reported, showing a good response after steroid treatment [197,198]. Post COVID-19 vaccination, otologic complications have been described and include hearing loss, tinnitus, dizziness, and vertigo [199]. Although vertigo was frequently suggested as a common neurological side effect after vaccination, only a few cases of vestibular neuropathy have been reported [200,201].

## 14. Other Peripheral Nervous System Disorders 

Parsonage–Turner syndrome or neuralgic amyotrophy is a rare neurological disorder typically characterized by the abrupt onset of unilateral shoulder pain followed by progressive brachial motor weakness [202]. The etiology of this disorder is unclear, but an inflammation of the brachial plexus is often evident. The syndrome has been associated with the post-surgery, post-traumatic, post-infectious, and post-vaccination state [203]. Several cases of Parsonage–Turner syndrome have been described following the COVID-19 vaccination [204]. The onset of the disorder has been generally reported within ten days from the vaccination, mostly hitting people in the third-to-fifth decade [205]. Neurophysiological and imaging tests sustain the diagnosis, showing altered brachial plexus nerve conduction studies and enlargement of the brachial plexus at the MRI, respectively [206]. As in previously described Parsonage-Turner cases, a good response to high-dose steroids has been reported [207,208]. There is no association with a specific vaccine. The largest Parsonage–Turner case series described twelve patients who developed the syndrome after one of the four major vaccines (five received Pfizer–BioNTech, four Oxford–AstraZeneca, two Janssen, and one had Moderna), generally subsequently to the first dose. Intriguingly the involved arm was homolateral to the site of vaccine injection [209].

Varicella-Zoster Virus (VZV) reactivation after COVID-19 vaccination is reported. Overall, VZV reactivation causing shingles may be due to cell-mediated immunity dysfunction as in aging, diabetes, physical and psychological stress, and, rarely, after vaccination [210]. Few studies described VZV reactivation in patients suffering from autoimmune, such as Sjogren’s syndrome, rheumatoid arthritis, systemic lupus erythematosus, and autoimmune inflammatory rheumatic disease, with recovery after antiviral therapy [211,212]. VZV reactivation has also been described in immunocompetent patients [213]. The seven cases described by Psichogiou and colleagues were older than 50 years, without risk factors for the immunosuppressive state, and they all received the Pfizer–BioNTech vaccine. The herpes zoster appeared seven to 20 days after the vaccination, and six out of seven patients recovered with valacyclovir treatment [214]. VZV reactivation is reported mostly after mRNA COVID-19 vaccination, especially with the Pfizer–BioNTech vaccine, and usually after the first dose [215]. However, the link between HZV reactivation and COVID-19 vaccination is not undeniable. In the Israelian observational historical cohort study conducted by Shasha and colleagues, the development of Herpes Zoster after COVID-19 vaccination occurred with a rate of 55.2 per 10,000 person-years vs. 51.5 cases per 10,000 person-years in the control unvaccinated group, thus showing a non-significant increase in the vaccinated population [190]. 

Few biopsy-proven small fiber neuropathy cases after receiving COVID-19 vaccination have been described. A 57-year-old female presented a subacute onset of intense, distal, burning dysesthesias after the second dose of the Pfizer vaccine, showing a good response to gabapentin symptomatic treatment [216]. A 43-year-old man developed neurosarcoidosis and small fiber neuropathy three days after the first dose of the Pfizer vaccine [93]. The development of autoantibodies targeting neuronal proteins via molecular mimicry has been proposed as the pathogenic mechanism for small fiber neuropathy, which has also been described after the Moderna vaccine [217].

## 15. Functional Neurological Disorders

The COVID-19 pandemic had a dramatic psychological impact. A higher rate of behavioral and neuropsychiatric disturbances has been described in the general population, healthcare workers, and people suffering from neurological conditions [218,219,220]. On the other hand, the psychological stress related to mass vaccination may trigger functional or psychogenic disorders [221]. Functional neurological disorders have been described after COVID-19 vaccination and are usually characterized by sudden onset, inconsistency over time, and normal diagnostic workup [222]. Anxiety-related events, feeling syncope, and dizziness are prevalent [223]. Functional neurological manifestations may be highly variable and include motor and sensory deficits mimicking strokes [224], episodic loss of consciousness and pseudo crisis [225], and bizarre movement disorders [226]. Multiple functional neurological disturbances, including bilateral facial palsy, hemiparesis, and facial hypoesthesia, have been described in a man as occurring after every single dose of vaccine [227]. The occurrence of psychogenic neurological disorders is a particularly delicate theme since they can devastate public opinion [10], feeding the debate against the COVID-19 mass vaccination and favoring the hesitancy phenomenon.

## 16. Discussion

In the vaccine human history, all kinds of vaccines have been associated with the possible appearance of neurological complications. COVID-19 vaccines are not free from neurological side effects, which have been reported since the first months of mass vaccination. 

Four major types of vaccines are at the moment available for the COVID-19 vaccines: DNA-based vaccines, mRNA-based vaccines, protein-based vaccines, and inactivated viruses. DNA-based vaccines use viral vectors to introduce the DNA coding for the spike protein. The mRNA vaccines use lipid nanoparticles to introduce mRNA into cells. Protein-based vaccines use the spike protein or its fragments. Lastly, other vaccines are based on inactivated virus [9]. 

Regarding the pathogenesis of COVD-19 vaccine-associated adverse events, several proposed mechanisms have been hypothesized based on the type of immunization. The mRNA vaccines play their action by inducing the encoding of viral antigens by the host cells. The elicited immune response stimulates CD4 and CD8 lymphocytes, eventually forming memory B cells. Viral vector vaccines are based on administering non-pathogenic viruses carrying antigen genes of the target (SARS-CoV-2) virus. Thus, the expression of the antigenic protein of the target virus induces the immune response [17,228]. Vaccines can cause general adverse events, which are strictly linked to the specific characteristics of the vaccine, and include local pain and edema, malaise, and tiredness. In addition, systemic and more severe events can be triggered by the vaccine-related immune response or associated with allergic reactions [17]. Human cells’ spike protein expression might trigger an inflammatory reaction leading to neurological autoimmune complications [82,85,89]. On the other hand, severe allergic reactions are rare but reported events since all vaccines (and the vaccine components) may induce anaphylaxis in some people [228].

Any patient or healthcare provider can report vaccine side effects through the Centers for Disease Control VAERS. However, a VAERS database limitation is that it is based on passive surveillance, therefore possibly reporting bias and errors [9]. Real-world data provided reports on a broad spectrum of severe neurological complications following COVID-19 vaccination. However, the real-world evidence is reassuring [229]. Large-scale epidemiological studies are needed to appropriately investigate the effective frequency of neurological complications after the COVID-19 vaccine.

Our review discusses the neurological side effects of COVID-19 vaccinations that are usually mild, short-duration, and self-limiting [230]. Severe unexpected post-vaccination complications that might occur due to molecular mimicry and subsequent neuronal damage are rare but possible events. Most severe neurological complications are reported in isolated case reports or small case series; therefore, a causal association between these adverse events is controversial. Establishing a clear relationship between the COVID-19 vaccine and autoimmune events is difficult. It is mandatory to have ongoing surveillance and reporting of adverse events associated with COVID-19 vaccines to ensure transparency concerning potential risks to patients. 

The fear of neurological complications fed the “vaccine hesitancy” [10]. COVID-19 vaccinations are usually well-tolerated, and the benefit for the global population outweighs the relatively rare mild-to-moderate side effects. In addition, the SARS-CoV-2 infection seems to be associated with an increased risk of neurological complications, which exceeds that of all the COVID-19 vaccines [192]. The risk of all neurological complications in the month after a SARS-CoV-2 infection is substantially higher than the risk of neurological side effects after COVID-19 vaccination [192]. 

The theme of vaccination remains of utmost importance since recent evidence confirmed the efficacy of the vaccine in preventing the COVID-19 infection but also suggests the waning of the immunity over time [231]. An important communication challenge remains, especially when considering individuals developing neurological complications and the impact of such effects on public opinion. Clinicians should discuss this occurrence with the patients and their families, especially prospecting new future pandemic waves. Vaccines are considered safe and effective drugs, but adverse effects are inevitable, especially during mass immunization, which allowed us to crush the pandemic.

## Data Availability

Not applicable.

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
