# Peer review of "Safety of COVID-19 Vaccines: Spotlight on Neurological Complications"

_life, 2022, doi:10.3390/life12091338_

Round 1
Reviewer 1 Report
I THINK IT IS A COMPREHENSIVELY WRITTEN REVIEW. IT CAN BE ACCEPTED.
Author Response
We would like to thank the reviewers for their careful reading of our paper and for the useful comments and suggestions, which helped us to improve the quality of the manuscript.
Our answers follow the reviewer’s comments, which are reported in italics. We tracked all the changes made to the manuscript, and updated the references.
Reviewer 1
I think it is a comprehensively written review. It can be accepted.
We thank the referee for the positive comment.
Reviewer 2 Report
I reviewed the manuscript entitled "Safety of COVID-19 vaccines: spotlight on neurological complications." The complications of COVID vaccination have been reported comprehensively; however, there are some remarks that should be considered: 1. Introduction: a) The definition of neurological complications and their sub-types needs to be introduced. b) lines 58 to 63 should be addressed in more detail. c) Vaccine type/platform should be considered when you are reporting the advantages and disadvantages. c) Neurological complications with other immunizations (mumps, ...) is also better to be introduced in a short paragraph. 2. a)The term "any of the approved vaccines" needs to be clarified. Approved by WHO, FDA, etc.; which one did you mean? b) "cerebral thrombotic complication" needs to be introduced and discussed more (e.g., which vaccine caused it, the mechanisms etc.) 3. a)VITT mechanisms need to be mentioned. b) "Conversely, patients with VITT do not have previous exposure to heparin ..." to the end of line 125 is hard to be understand, and needs to be reported in an organized, simple form. c) Lack of cohesion and coherence in this part made it difficult to follow the section. It might be better to divide it to some sub-types. 4. a) Some general information has been mentioned in this part, which is not clear that is refeered to which vaccine, so it needs to be mentioned in detail. (e.g., In the past, vaccines were awarded as a possible determinant...) b) A brief history of other vaccine complications is better to be added. c) The lines 212 to 220 cut the integration of this section, so some connections need to be added to resolve this problem. 5. As it is mentioned in the manuscript, many different studies reported "Myelitis" as an adverse effect of COVID-19 vaccination. It has been suggested to provide a table to review all of these papers ( Some variables like Vaccine type, clinical status, severity, treatment strategy, etc. can be extracted in this table) 6. "Optic neuritis" associated reports should be reported in a clear order ( oldest to newest, case reports to trials, etc.); this point should be considered in the other sections. 7. "Regarding the pathogenesis of COVD-19 vaccine-associated adverse events, several proposed mechanisms have been hypothesized based on the type of immunization" needs to be discussed more and compare the possible mechanisms.
Author Response
We would like to thank the reviewers for their careful reading of our paper and for the useful comments and suggestions, which helped us to improve the quality of the manuscript.
Our answers follow the reviewer’s comments, which are reported in italics. We tracked all the changes made to the manuscript, and updated the references.
Reviewer 2
I reviewed the manuscript entitled "Safety of COVID-19 vaccines: spotlight on neurological complications." The complications of COVID vaccination have been reported comprehensively; however, there are some remarks that should be considered:
-
Introduction: a) The definition of neurological complications and their sub-types needs to be introduced. b) lines 58 to 63 should be addressed in more detail. c) Vaccine type/platform should be considered when you are reporting the advantages and disadvantages. c) Neurological complications with other immunizations (mumps, ...) is also better to be introduced in a short paragraph.
We thank the referee for the comments and the suggestions. We improved the introduction section, further explaining neurological complications and their subtypes. We added a more detailed description of significant studies on side effects of COVID-19 vaccination. We discussed vaccine types and complications (lines 81-88), and we added a short paragraph on the neurological adverse effects of other vaccines (lines 89-96).
-
a)The term "any of the approved vaccines" needs to be clarified. Approved by WHO, FDA, etc.; which one did you mean? b) "cerebral thrombotic complication" needs to be introduced and discussed more (e.g., which vaccine caused it, the mechanisms etc.)
We thank the referee for the comment. We clarified the underlined sentence about headache in FDA and EMA approved-vaccines. We added a more detailed explanation of the cited study by García-Azorín and colleagues; we discussed in-depth CVST in the subsequent paragraph.
-
a)VITT mechanisms need to be mentioned. b) "Conversely, patients with VITT do not have previous exposure to heparin ..." to the end of line 125 is hard to be understand, and needs to be reported in an organized, simple form. c) Lack of cohesion and coherence in this part made it difficult to follow the section. It might be better to divide it to some sub-types.
We thank the referee for the comment and suggestions. We agree and we amended the text accordingly, reformulating the sentence about similarities between VITT and HIT (lines 140-142); in addition, we divided paragraph 3 in two subsections, to add clarity (3.1 Vaccine-Induced Thrombotic Thrombocytopenia and Cerebral Venous Sinus Thrombosis and 3.2 Ischemic and hemorrhagic strokes)
-
a) Some general information has been mentioned in this part, which is not clear that is refeered to which vaccine, so it needs to be mentioned in detail. (e.g., In the past, vaccines were awarded as a possible determinant...) b) A brief history of other vaccine complications is better to be added. c) The lines 212 to 220 cut the integration of this section, so some connections need to be added to resolve this problem.
We thank the referee for the comment. We added the specific vaccine that was accounted as possible determinant of MS, and we better specified the possible risks of vaccination in the MS population. We agree with the referee’s last comment, since some information was missing in the original draft to understand the sub-section clearly, we now added more info, and we feel like now all is more understandable (lines 267-274).
-
As it is mentioned in the manuscript, many different studies reported "Myelitis" as an adverse effect of COVID-19 vaccination. It has been suggested to provide a table to review all of these papers (Some variables like Vaccine type, clinical status, severity, treatment strategy, etc. can be extracted in this table)
We thank the referee for the suggestion. However, we believe adding a table only for “myelitis” as a neurological complication among all the presented in the review might unbalance the review. In addition, providing a systematic revision of all the reported cases of myelitis is beyond the scope of this narrative review, and we feel that a table for the described publications may be incomplete. We thus decided not to add to the manuscript any table, favoring a more fluid discussion of the literature.
-
"Optic neuritis" associated reports should be reported in a clear order (oldest to newest, case reports to trials, etc.); this point should be considered in the other sections.
We thank the reviewer for the interesting comment, we fully agreed and this section was reported as requested in a clearer order: oldest to newest.
-
"Regarding the pathogenesis of COVD-19 vaccine-associated adverse events, several proposed mechanisms have been hypothesized based on the type of immunization" needs to be discussed more and compare the possible mechanisms.
We thank the referee for the comment. We agreed and we amended the text accordingly, adding a more comprehensive discussion about vaccine types and mechanisms.
Round 2
Reviewer 2 Report
The authors addressed all of my comments. Thanks.